# COMBINATION OF SUPERVISED AND REINFORCEMENT LEARNING FOR VISION-BASED AUTONOMOUS CONTROL

## ABSTRACT

Reinforcement learning methods have recently achieved impressive results on a wide range of control problems. However, especially with complex inputs, they still require an extensive amount of training data in order to converge to a meaningful solution. This limitation largely prohibits their usage for complex input spaces such as video signals, and it is still impossible to use it for a number of complex problems in a real world environments, including many of those for video based control. Supervised learning, on the contrary, is capable of learning on a relatively small number of samples, however it does not take into account reward-based control policies and is not capable to provide independent control policies. In this article we propose a model-free control method, which uses a combination of reinforcement and supervised learning for autonomous control and paves the way towards policy based control in real world environments. We use Speed-Dreams/TORCS video game to demonstrate that our approach requires much less samples (hundreds of thousands against millions or tens of millions) comparing to the state-of-the-art reinforcement learning techniques on similar data, and at the same time overcomes both supervised and reinforcement learning approaches in terms of quality. Additionally, we demonstrate the applicability of the method to MuJoCo control problems.

## 1 INTRODUCTION

Recently proposed reinforcement learning methods (Mnih et al. (2013), Kober & Peters (2012), Lillicrap et al. (2015)) allowed to achieve significant improvements in solving complex control problems such as vision-based control. They can define control policies from scratch, without mimicking any other control signals. Instead, reward functions are used to formulate criteria for control behaviour optimisation. However, the problem with these methods is that in order to explore the control space and find the appropriate actions, even the most recent state-of-the-art methods require several tens of millions of steps before their convergence (Mnih et al. (2013), Mnih et al. (2016)).

One of the first attempts to combine reinforcement and supervised learning can be attributed to Rosenstein et al. (2004). Recently proposed supervised methods for continious control problems, like Welschehold et al. (2016), are known to rapidly converge, however they do not provide independent policies. Contrary to many classification or regression problems, where there exists a definitive labelling, as it is presented in Szegedy et al. (2017), Krizhevsky et al. (2012), for many control problems an infinitely large number of control policies may be appropriate. In autonomous driving problems, for example, there is no single correct steering angle for every moment of time as it is appropriate to follow different trajectories; however such control sequences can be assessed by their average speeds, or time of the lap, or other criteria appropriate for some reason. The situation is the same for other control problems connected with robotics, including walking (Li & Wen (2017)) and balancing (Esmaeili et al. (2017)) robots, as well as in many others (Heess et al. (2017)). In these problems, also usually exist some criteria for assessment (for example, time spent to pass the challenge), which would help to assess how desirable these control actions are.

The problem becomes even more challenging if the results are dependent on the sequence of previous observations (Heess et al. (2015)), e.g. because of dynamic nature of the problem involving speed or acceleration, or the difference between the current and the previous control signal.

In many real-world problems, it is possible to combine the reinforcement and the supervised learning. For the problem of autonomous driving, it is often possible to provide parallel signals of the autopilot in order to use this information to restrict the reinforcement learning solutions towards the sensible subsets of control actions. Similar things can also be done for robotic control. Such real world models can be analytical, or trained by machine learning techniques, and may use some other sensors, which are capable to provide alternative information (e.g., the model trained on LiDAR data can be used to train the vision based model). However, although there were some works using partially labelled datasets within the reinforcement learning framework (Večerík et al. (2017)), as far as we believe, the proposed problem statement, injecting supervised data into reinforcement learning using regularisation of $Q$-functions, is different from the ones published before. In Večerík et al. (2017), the authors consider the problem of robotic control which does not involve video data, and their approach considers sharing the replay buffer between the reinforcement learning and demonstrator data.

The novelty of the approach, presented in this paper, is given as follows:

1. the regularised optimisation problem statement, combining reinforcement and supervised learning, is proposed;

2. the training algorithm for the control method is proposed, based on this problem statement, and assessed on the control problems;

3. the novel greedy actor-critic reinforcement learning algorithm is proposed as a part of the training algorithm, containing interlaced data collection, critic and actor update stages.

The proposed method reduces the number of samples from millions or tens of millions, required to train the reinforcement learning model on visual data, to just hundreds of thousands, and also improves the quality against the supervised and reinforcement learning.

## 2 PROPOSED METHOD

The overall idea of the method is shown in figure 1 and can be described as follows: to perform an initial approximation by supervised learning and then, using both explicit labels and rewards, to fine-tune it.

For supervised pre-training, the annotated dataset should contain recorded examples of control by some existing model. The aim of this stage is to mimic the existing control methods in order to avoid nonsense control behaviour of the trained model.

For supervised and reinforcement learning based fine-tuning, a pretrained model is used as an initial approximation. Contrary to the standard reinforcement learning approaches, the pre-trained model helps it to avoid nonsense control values right from the beginning. Also, for the control it is assumed that there is access to labels, which helps to divert the reinforcement learning model from spending its time on exploring those combinations of inputs and control signals, which are most likely not to provide meaningful solutions.

### 2.1 SUPERVISED PRETRAINING

Hereinafter we consider the following problem statement for supervised pretraining. Let $Z$ be the space of all possible input signals. To simplify the formalisation, we restrict ourselves to the image signals $z \in Z = \mathbb{R}^{m \times n}$, where $m$ and $n$ are the height and the width of the image, respectively. Such signals can form sequences of finite length $l > 0 : \langle z_1, z_2, \ldots, z_l \rangle \in Z^l$. We define an operator from the subsequences of real valued quantities to $d$-dimensional control signals $\pi(z_i, z_{i-1}, \ldots, z_{i-p+1}|\Theta_\pi) : Z^p \to \mathbb{R}^d$, where $i \in [p, l]$, $p \in \mathbb{N}$ is the number of frames used to produce the control signal, and $\Theta_\pi$ are the parameters of the operator $\pi(\cdot)$. We denote $c_i = \pi(z_i, \ldots, z_{i-p+1}|\Theta_\pi), x_i = (z_i, z_{i-1}, \ldots, z_{i-p+1})$.

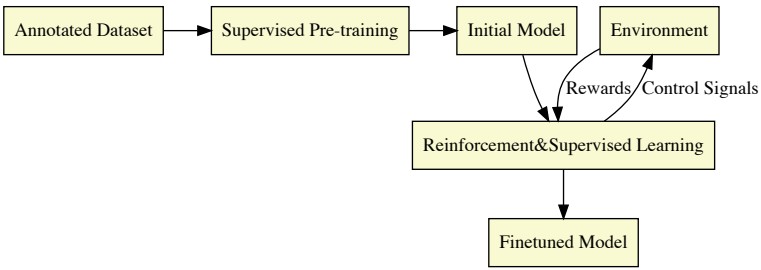

Figure 1: The overall scheme of the proposed method

The problem is stated as follows: given the set of $N$ sequences $\left\{ \hat{z}^j \in Z^{l_j} \right\}_{j=1}^{N}$ and the set of corresponding signal sequences $\left\{ \left\langle \hat{c}_i^j \in \mathbb{R}^d \right\rangle_{i=1}^{l_j} \right\}_{j=1}^{N}$, produced by some external control method called a 'reference actor', find the parameters $\Theta_\pi$ of the actor $\pi(\cdot | \Theta_\pi)$, which minimise a loss function (the used loss function is defined in formula (7)).

## 2.2 LABEL-ASSISTED REINFORCEMENT LEARNING

The reinforcement learning method is inspired by the DDPG algorithm (Lillicrap et al. (2015)), however, it is substantially reworked in order to meet the needs of the proposed combination of supervised and reinforcement learning working in real time. First, the problem statement and basic definitions, necessary for formalisation of the method, need to be given.

In line with the standard terminology for reinforcement learning, we refer to a model, generating control signals, as an agent, and to control signals as actions. Also, as it is usually done for reinforcement learning problem statements, we assume the states to be equivalent to the observations. Initially, the model receives an initial state of the environment $x_1$. Then it applies the action $c_1$, which results in transition into the state $x_2$. The procedure repeats recurrently, resulting in sequences of states $X^n = \langle x_1, x_2, \ldots, x_n \rangle$ and actions $C^n = \langle c_1, c_2, \ldots, c_n \rangle$. Every time the agents performs an action it receives a reward $r(x_i, c_i) \in \mathbb{R}$ (Lillicrap et al. (2015)).

The emitted actions are defined by the policy $\pi$, mapping states into actions. In general (Silver et al. (2014)), it can be stochastic, defining a probability distribution over the action space, however here the deterministic policy is assumed. For such a deterministic policy, one can express the discounted future reward using the following form of the Bellman equation for the expectation of discounted future reward $Q(\cdot, \cdot)$ (Lillicrap et al. (2015)):

$$Q(x_i, c_i) = \mathbb{E}_{x_{i+1}} \left[ r(x_i, c_i) + \gamma Q(x_{i+1}, \pi(x_{i+1})) \right], \tag{1}$$

where $\pi(\cdot)$ is an operator from states to actions, $\gamma \in [0, 1]$ is a discount factor.

$Q(\cdot, \cdot)$ is approximated by the critic neural network in line with the actor-critic approach. Similarly to Lillicrap et al. (2015), the proposed method uses a replay buffer to provide a training set.

In many of the state-of-the-art works on reinforcement learning, the actor's parameters are trained by maximisation of the critic's approximation of $Q(\cdot, \cdot)$, using gradient descent with gradients, obtained using the chain rule (Lillicrap et al. (2015), Silver et al. (2014)):

$$\nabla_{\Theta_\pi} \pi(x | \Theta_\pi) \approx \mathbb{E}_\pi [\nabla_{\pi(x)} Q(x, \pi(x) | \Theta_Q) \nabla_{\Theta_\pi} \pi(x | \Theta_\pi)], \tag{2}$$

where $\Theta_Q$ and $\Theta_\pi$ are the trainable parameters of the actor and the critic, respectively. In Lillicrap et al. (2015), Silver et al. (2014), the greedy optimisation algorithm is not used as it might be impossible to collect diverse training sample for the critic as the actor would not be able to advance through the task. Instead, small steps in the gradient direction are applied to slightly modify the policy.

In the proposed method, contrary to many state-of-the-art methods, the optimisation is carried out in a greedy way, so that the steps of testing the current policy, updating the critic and the actor are

interlaced. This is done in order to meet the requirements of real world scenarios, namely minimising the difference between the measurements per second rate in testing and training scenarios and providing the reasonable performance of the actor after the smallest possible number of epochs. In order to avoid deterioration of performance in the pretrained model (which would essentially lead to the number of steps comparable with the state-of-the-art reinforcement learning models), the regularisation is used to bring the parameters closer towards some pre-defined (reference) policy.

In the following derivations we use a restriction that $Q(\cdot, \cdot) \geq 0$, which, as one can easily see from Equation (1), will be true if the rewards are non-negative. We also assume (due to the practical reasons, it doesn't affect further derivations) that the control signal is bounded between values $(t_1, t_2)$, and $t_1$ and $t_2$ are not included into the appropriate operational values. Based on these practical assumptions, the training sample for the critic is augmented with the values $Q(x, t_1) = Q(x, t_2) = 0$ for every value $x$.

The optimisation problem is based on the regularised $Q$-function $f(x, \pi(x|\Theta_\pi))$ :

$$F(\Theta_\pi) = \sum_{x \in X} f(x, \pi(x|\Theta_\pi)) \to \max_{\Theta_\pi}, \tag{3}$$

$$f(x, \pi(x|\Theta_\pi)) = Q(x, \pi(x|\Theta_\pi)) - \alpha Q(x, \hat{\pi}(x)) \times \rho(\pi(x|\Theta_\pi), \hat{\pi}(x)), \tag{4}$$

where $\alpha \geq 0$ is some coefficient, $\hat{\pi}(\cdot)$ is the reference actor, and $\rho(\cdot, \cdot)$ is a differentiable distance-like function. One can easily see that in the case $\alpha = 0$ it completely coincides with the reinforcement problem, and with $\alpha \to \infty$ the problem becomes a standard supervised training one. The weight $Q(x, \hat{\pi}(x))$ is given to encourage the actor to follow the reference policy if the expected reward is high, and not to encourage much otherwise.

After differentiating this expression with respect to $\Theta_\pi$ one can see that

$$\nabla F(\Theta_\pi) = \sum_{x \in X} \nabla_{\Theta_\pi} \pi(x|\Theta_\pi) \left\{ \nabla_\pi Q(x, \pi(x|\Theta_\pi)) - \alpha Q(x, \hat{\pi}(x)) \frac{\partial \rho(\pi(x|\Theta_\pi), \hat{\pi}(x))}{\partial \pi} \right\}. \tag{5}$$

As in Lillicrap et al. (2015), the update of the critic is carried out by solving the following optimisation problem:

$$\sum_{(x_i, c_i) \in (\hat{X}, \hat{C})} \left[ \hat{Q}^\pi(x_i, c_i|\Theta_Q) - r(x_i, c_i) - \gamma \hat{Q}^\pi(x_{i+1}, \pi(x_{i+1}|\Theta_\pi)|\Theta_Q) \right]^2 \to \min_{\Theta_Q}, \tag{6}$$

where $\hat{X}$ and $\hat{C}$ are taken from the replay buffer, $\hat{Q}^\pi$ is the discounted reward function approximation. One can see that this equation is recurrent, and therefore the current target for training is dependent on the values of the previous training stage.

The training procedure is carried out in the way described in Algorithm 1. In this algorithm, NUM_EPOCHS denotes the maximum number of epochs during the training procedure.

In a contrast to Lillicrap et al. (2015), the proposed model detaches the procedures of filling the replay buffer, training critic and actor into three subsequent steps, making the algorithm greedy rather than making steps every time control happens. As one can see from the algorithm, the 0-th epoch's testing episodes reflect the performance of the model with supervised pretraining. This was done to assess the performance of the same model parameterisation during the epoch, as well as to exclude the problem when the control signal frequency (and hence performance) is affected by the optimisation time.

## 3 RESULTS

To demonstrate the ability of the method to learn control signals, TORCS (Espi et al.)/ SpeedDreams (SpeedDreams) driving simulator was used. As a reference actor and for the baseline, the Simplix bot, which is a part of the standard package, was used. The outputs of the bot steering angles were restricted between $-0.25$ and $0.25$. The car name was Lynx 220. In the case if the car has stuck (average speed is less than one unit as measured by the simulator), the recovery procedure is provided by the bot. The time and rewards for the recovery are excluded from consideration. The reward is defined as the current car speed measured by the simulator. The assessment has been carried out on a single computer, using NVIDIA GeForce GTX 980 Ti graphical processor on Ubuntu Linux operating system. The model parameters are described in the Appendix A.

---

**Algorithm 1** The description of the training algorithm

Initialise and train the parameters $\Theta_\pi$ for the actor $\pi(x|\Theta_\pi)$ on a supervised dataset with observations and labels $\{X_S, C_S\}$
Initialise the parameters of the critic $\Theta_Q$, $k = 1$
Initialise the empty replay buffer $B$
**while** $k \leq$ NUM_EPOCHS **do**

  Perform N_TESTING_EPISODES testing episodes of the length L_TESTING_EPISODES with the current parameters $\Theta_\pi$, where the states $x_i$, actions $c_i$, reference actor actions $c_i^{GT}$, rewards $r_i = r(x_i, c_i)$ and subsequent states $x_{i+1}$ are collected and put into the replay buffer $B$

  Perform N_GD_UPDATE_CRITIC iterations of gradient descent, according to the optimisation problem (6), in order to update the parameters of the critic $\Theta_Q$; values $x_i, x_{i+1}, c_i, r_i$ are taken from the replay buffer $B$

  Perform N_GD_UPDATE_ACTOR iterations of gradient descent for the optimisation problem (3) in order to update the parameters of the actor $\Theta_\pi$; values $x_i, c_i^{GT} = \hat{\pi}(x_i)$ are taken from the replay buffer $B$

  $k = k + 1$
**end while**

---

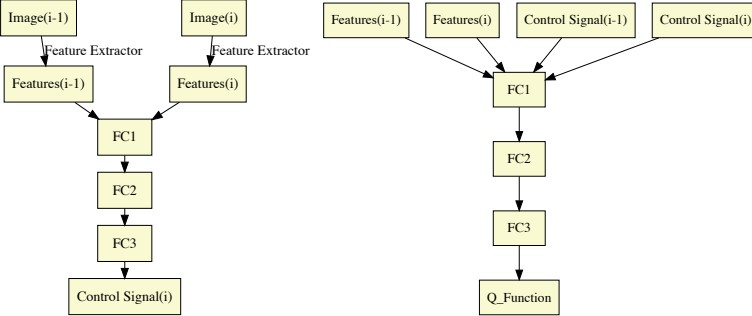

Figure 2: The scheme of the actor (left) and critic (right) networks

## 3.1 NETWORK ARCHITECTURE

The network architecture, used for the proposed actor-critic approach, is shown in Figure 2. For supervised pretraining, the dataset has been collected from the SpeedDreams simulator, combining the sequences of images and the corresponding control values produced by the bot. Using this dataset, the InceptionV1 network (Szegedy et al. (2015)) is first fine-tuned from the Tensorflow Slim implementation (Silberman & Guadarrama (2016)) on the collected dataset, mapping each single image directly into the control signals. The last (classification) layer of the network is replaced by a fully connected layer of the same size (1001) as the preceding logits layer; this layer is referred to as a feature extraction layer as it serves as an input space for the actor-critic model. Such an approach is used in order to avoid expensive storage of images in the replay buffer, as well as circumvent the challenge of training deep architectures within the reinforcement learning setting. Each of the actor and critic layers except the last contains ReLU nonlinearity (Nair & Hinton (2010)). The actor's last layer with dimensionality $d$ of the control signal is followed by the $\tanh$ nonlinearity, which restricts the output between the boundary values $(t_1, t_2) = (-1, 1)$; the last layer of the critic has no nonlinearities.

For supervised finetuning and pretraining stages, we use the following loss function:

$$l(X, C) = \sum_{x \in X, c \in C} \frac{|\pi(x) - c|}{|c| + \epsilon}, \tag{7}$$

where $\epsilon$ is a reasonably small constant (with respect to $c$, we use $\epsilon = 10^{-2}$), $\pi(x)$ is an operator, transforming input vectors $x$ to the control signals, $c$ is the reference actor control signal. This was done because some of the control signals, supplied by the bot, were large enough to devaluate the impact of the smaller values within the loss.

Table 1: Simplix bot rewards

| Test episode 1 | Test episode 2 | Test episode 3 | Test episode 4 | Test episode 5 | Average |
|---|---|---|---|---|---|
| 81315.25 | 81779.02 | 79481.26 | 81563.64 | 81931.15 | 81214.07 |

Table 2: Maximum sum-of-rewards values for different parameters $\alpha$ (for $\alpha = 0$ these values are obtained during supervised pretraining)

| $\alpha$ | $\max_R$ | $100\% \cdot \frac{\max_R}{\max_{R_{bot}}}$ | Epoch | $\max_{\overline{R}}$ | $100\% \cdot \frac{\max_{\overline{R}}}{\overline{R}_{bot}}$ | Epoch |
|---|---|---|---|---|---|---|
| 100 | 75511.69 | 92.16 | 9 | 72180.06 | 88.88 | 10 |
| 0.25 | 75896.42 | **92.63** | 17 | 73339.50 | 90.30 | 14 |
| 0.1 | 75866.78 | 92.60 | 32 | 73380.70 | **90.35** | 15 |
| 0.05 | 75379.34 | 91.97 | 27 | 71670.40 | 88.25 | 23 |
| 0.01 | 67766.42 | 82.71 | 20 | 65806.41 | 81.03 | 20 |
| 0 | 62724.68 | 76.56 | 1 | 58469.25 | 71.99 | 1 |

Similarly, the distance-like function $\rho$ in formula (4) is defined as:

$$\rho(\pi, \hat{\pi}) = \frac{(\pi - \hat{\pi})^T (\pi - \hat{\pi})}{|\hat{\pi}|^2 + \epsilon}, \tag{8}$$

where $\epsilon$ is a reasonably small constant (we use $\epsilon = 10^{-4}$).

To implement the formalisation, described in section 2.1, we use a syamese network architecture (see Bertinetto et al. (2016, October)), given in Figure 2. The features, calculated by the fine-tuned network, are submitted for the current and the previous frame ($p = 2$ in terminology of section 2.1). It is done in order to take into account the dynamic state of the environment, including speed. Also we need to mention that the previous control signal is submitted as an input for the critic (for the initial video frame in each sequence, we assume that the frames and control signals are the same for the current and the previous frames).

## 3.2 EXPERIMENTS

Table 2 and Figure 3 contain the comparison of the results given different values of the parameter $\alpha$. When $\alpha = 0$, we rely solely on reinforcement learning; the higher is the parameter $\alpha$, the more restriction is put to stick to the supervised labels rather than improving the policy by reinforcement learning.

In Figure 3, the scatter points depict total rewards during each of the testing episodes, and the curves are depicting the mean total rewards for each epoch over all testing episode rewards. The total reward is calculated as an arithmetic sum of the rewards during one training episode. The left figure shows the results for all tested parameters, while the right one shows the comparison between the best performing one and the one with the largest value of $\alpha$ (i.e. the closest to supervised active learning). The performance of the pretrained model corresponds to the first epoch in the graph. The shaded area in the right figure shows the standard deviation of the testing episodes performance during the epoch. One can see from the figure that the smaller values of the coefficient $\alpha$ tend to yield worse performance; however, it may be attributed to longer convergence time as it implies more reliance on reinforcement learning exploration. At the same time, the unlimited increase of the coefficient $\alpha$ does not help gaining further performance improvements. Even more, one can see from the right figure that after certain point the curve for $\alpha = 100$ slowly declines; we suggest that it could be caused by overfitting to the bot values. We also see that for $\alpha = 0$, when the supervised data is used only during the pre-training stage, the performance is much lower than for the rest of the graphs.

Table 2 shows the total rewards for different values of parameter $\alpha$. The value $\max_R$ shows the maximum total reward over one testing episode during all the training time (corresponds to the highest scatter point in Figure 3 for a given parameter $\alpha$), the value $\max_{\overline{R}}$ shows the maximum

mean total reward, where the mean is calculated for each epoch over all testing episode total rewards (corresponds to the highest point of the curve in Figure 3 for a given parameter $\alpha$).

In order to compare the rewards shown in these graphs and tables, we have also measured them the Simplix bot in the same conditions (frame per second rate) as the proposed algorithm. The total rewards for the Simplix bot, available in the SpeedDreams/TORCS simulator, are given in Table 1. The mean value of these rewards is $\overline{R_{bot}} = 81214.07$. The maximum value achieved by the bot is $\max_{R_{bot}} = 81931.15$. The percentage of the proposed algorithm's rewards with respect to the bot one, in average and for the best performance, is also given in Table 2.

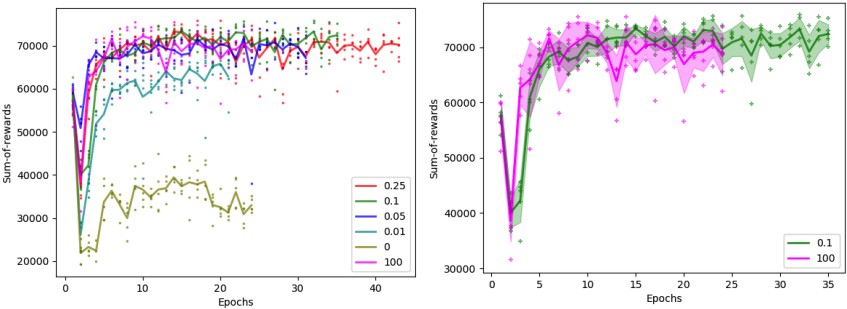

Figure 3: Sum-of-rewards for different parameters $\alpha$

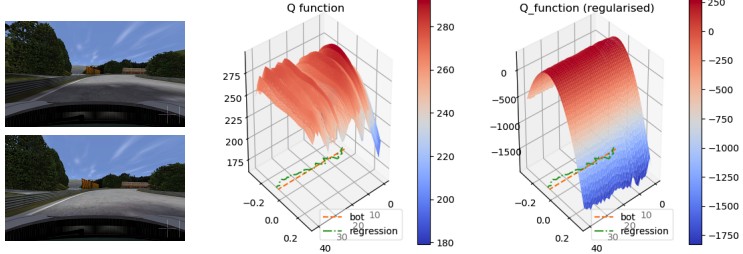

Figure 4: Regularised and non-regularised $Q$-function values, high reward, $\alpha = 0.25$; the epochs' indices are shown on the right axis, and the values of $\pi(\cdot)$ are shown on the left axis

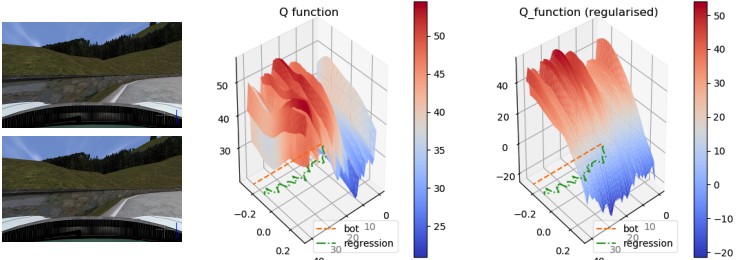

Figure 5: Regularised and non-regularised $Q$-function values, low reward, $\alpha = 0.25$; the epochs' indices are shown on the right axis, and the values of $\pi(\cdot)$ are shown on the left axis

One of the notable things is the dramatic decrease in the number of samples: from millions (Lillicrap et al. (2015)) or tens of millions (Mnih et al. (2013)) measurements for standard reinforcement learning techniques (Lillicrap et al. (2015)) to just hundreds of thousands (15000 per training epoch, several tens of training epochs). For some of the reinforcement learning algorithms, trained on a problem of driving in a simulator, only some realisations were able to finish the task (Lillicrap et al. (2015)), and for those methods, which report solving similar problems by reinforcement learning only, the reported performance constitutes $75 - 90\%$ (Mnih et al. (2016)), while we achieve up to $92.63\%$ of the bot's performance as reported in Table 2. The graphs in Figures 4 and 5 illustrate the difference between the unregularised $Q$-function $Q(x, \pi(x))$ and its regularised counterpart $f(x, \pi(x))$ (see Equation 4). The images on the top of the figures, recorded during the testing

episode of the first epoch (directly after the supervised pretraining), have been used for computation of the $Q$-values. The graphs below these images show the evolution of the $Q$-function (and its regularised version $f$) across the epochs. One can see that, as expected, the regularised $Q$-function is steeper than the unregularised one as it penalises the values which are too far away from the reference actor control signal. The difference between Figures 4 and 5 shows the importance of the factor $Q(x, \hat{\pi}(x))$ in Equation 4: the penalty for not following the control signal is less, if the expected reward is smaller. It makes the algorithm explore values further from the bot's values, if it does not provide high expected reward.

Another interesting aspect to be considered is how the proposed method would behave for other well-known control scenarios. Appendix B discusses the performance of the method for MuJoCo environments (Todorov et al. (2012)) using OpenAI Baselines package (Dhariwal et al. (2017)).

## 4    CONCLUSION

The proposed method shows dramatic improvement in the number of samples for video data (down to just several hundred thousand) comparing to the reinforcement learning methods, as well as improves the performance comparing to both supervised and reinforcement learning. We believe that such approach, combining reinforcement and supervised learning, could help to succeed in the areas of complex spaces where the state-of-the-art reinforcement learning methods are not working yet, as well as towards practical usage for real world models such as autonomous cars or robots.

However, there are still a few limitations of the proposed method. First, it still requires label data through all the course of training. We believe that in the future work it should be possible to reduce usage of training data to a limited number of labelled episodes. Such decrease of the training data could benefit to the range of practical tasks solvable by the proposed approach.

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

Table 3: Parameters of the network

| Parameter | Name in Algorithm 1 | Value |
|---|---|---|
| Critic Training Steps/ Epoch | N_GD_UPDATE_CRITIC | 12500 |
| Actor Training Steps/ Epoch | N_GD_UPDATE_ACTOR | 12500 |
| Testing episode length | L_TESTING_EPISODES | 3000 |
| Testing episodes per epoch | N_TESTING_EPISODES | 5 |
| Actor learning rate | | $1 \cdot 10^{-4}$ |
| Critic learning rate | | $1 \cdot 10^{-5}$ |
| Supervised learning rate | | $1 \cdot 10^{-4}$ |
| Supervised momentum | | 0.9 |
| Soft update coefficient | | 0.1 |
| Discount factor $\gamma$ | See Equation 1 | 0.9 |
| Actor FC1 size | See Figure 2 | 2002 |
| Actor FC2 size | See Figure 2 | 2002 |
| Actor FC3 size | See Figure 2 | 1 |
| Critic FC1 size | See Figure 2 | 2005 |
| Critic FC2 size | See Figure 2 | 2005 |
| Critic FC3 size | See Figure 2 | 1 |

## A    PARAMETERISATION OF THE MODEL

The parameterisation for the experiments is given in Table 3; the parameters' verbal description is augmented with the names referencing to Algorithm 1.

For supervised-only pretraining of the actor network, the Momentum algorithm is used (Qian (1999)); for the rest of the stages, the Adam algorithm is used (Kingma & Ba (2014)). The proposed algorithm has been implemented in Python using TensorFlow framework (Abadi et al. (2016)). For the stage of supervised pretraining, in order to improve convergence of the model at the initial stage, the additional soft update coefficient was introduced for exponential smoothing of the parameters of the network during gradient descent optimisation.

## B    EXPERIMENTS WITH REGULARISED DDPG ON MUJOCO ENVIRONMENTS

The experiments with MuJoCo environment (Todorov et al. (2012)) demonstrate the applicability of the proposed idea of regularising $Q$-values with supervised learning to other well known control problems. For this purpose, we have added the proposed $Q$-function regularisation to the standard DDPG algorithm with the implementation, network architecture and parameters, taken from OpenAI Baselines (Dhariwal et al. (2017)). $L^2$ distance has been used as $\rho$ for the equation (4). Also, in order to maintain the correctness of regularisation assumptions, as the condition $Q \geq 0$ is not met for some low reward values in MuJoCo, $Q(x, \hat{\pi}(x))$ was substituted by $\max(0, Q(x, \hat{\pi}(x))$. The reference actors were obtained by pretraining actors by standard DDPG algorithm.

The results of these experiments are given in Figure 6. In every graph, the black lines show the performance of the pretrained reference actor. These experiments are aimed to compare the following three cases:

1. the original DDPG algorithm (referenced as $\alpha = 0$)

2. the DDPG algorithm with fixed regularisation coefficient, $\alpha = 0.1$

3. the DDPG algorithm with exponential decay, initial value of $\alpha$ is 1, the value is decayed with the coefficient 0.01 every $20,000$ timesteps.

For the HalfCheetah scenario, one can see that the model with the fixed regularisation coefficient can easily reach the pretrained value but then lags behind the algorithm with exponential decay. The exponential decay algorithm, in contrary, takes advantage of the reference actor performance and then gradual decay of the regularisation enables it to explore values further from the reference actor.

These results could suggest that in certain cases the regularisation can prevent the model of further exploration beyond the reference actor performance.

For the Hopper scenario, the peak in the performance of original DDPG algorithm beyond the reference actor baseline near step 270000 suggests that the original DDPG algorithm may be unstable for this task, which also holds for DDPG with exponential decay as the regularisation coefficient becomes negligibly small. At the same time one can see that the model with the fixed regularisation coefficient can reach performance beyond the reference actor.

It could be concluded from the graphs for the InvertedDoublePendulum task that convergence depends in this case on the initial value of parameter $\alpha$. The larger initial value for the DDPG with exponential decay appears to give better results due to heavier reliance on the supervised part. What is important is that all versions of the method are able to maintain stable performance after the initial training episode.

For the Swimmer scenario, the exponential $\alpha$ setting allowed to go for some period of time beyond the reference actor baseline; at the same time, the version with $\alpha = 0.1$, while not beating the reference actor, shows smaller variance than the original algorithm, and most of the time exposes better average performance than the original DDPG method.

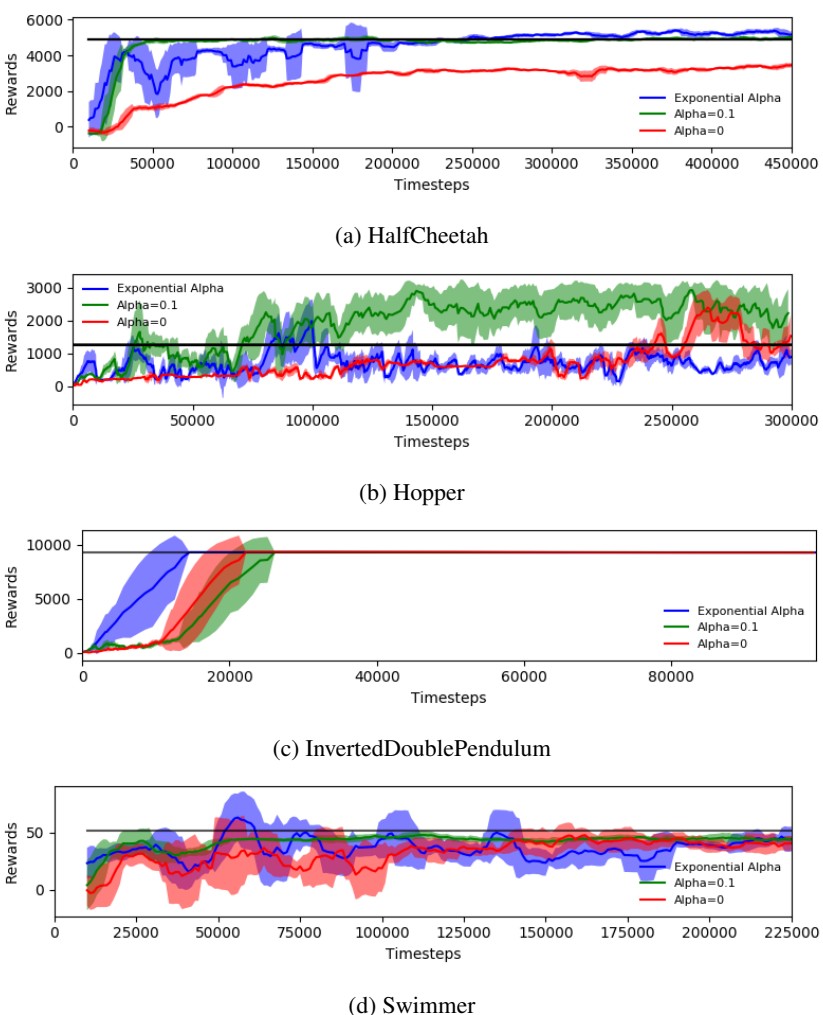

(a) HalfCheetah

(b) Hopper

(c) InvertedDoublePendulum

(d) Swimmer

Figure 6: Results in MuJoCo environments, top to bottom: HalfCheetah, Hopper, InvertedDoublePendulum, Swimmer. The average reference actor performance is shown by black horizontal bar, $\alpha = 0$ corresponds to the original DDPG algorithm

