# OpenReview forum: "Combination of Supervised and Reinforcement Learning For Vision-Based Autonomous Control"
_ICLR.cc/2018/Conference — Reject_

### Official Review · AnonReviewer1 · 2017-11-21
**Good ideas but needs more supporting evidence.**

**Rating:** 4
**Confidence:** 5

**Review:**

This paper proposes leveraging labelled controlled data to accelerate reinforcement-based learning of a control policy.  It provides two main contributions: pre-training the policy network of a DDPG agent in a supervised manner so that it begins in reasonable state-action distribution and regalurizing the Q-updates of the q-network to be biased towards existing actions.  The authors use the TORCS enviroment to demonstrate the performance of their method both in final cumulative return of the policy and speed of learning.

This paper is easy to understand but has a couple shortcomings and some fatal (but reparable) flaws:.

1) When using RL please try to standardize your notation to that used by the community, it makes things much easier to read.  I would strongly suggest avoiding your notation a(x|\Theta) and using \pi(x) (subscripting theta or making conditional is somewhat less important).  Your a(.) function seems to be the policy here, which is invariable denoted \pi in the RL literature.  There has been recent effort to clean up RL notation which is presented here: https://sites.ualberta.ca/~szepesva/papers/RLAlgsInMDPs.pdf. You have no obligation to use this notation but it does make reading of your paper much easier on others in the community.  This is more of a shortcoming than a fundamental issue.

2) More fatally, you have failed to compare your algorithm's performance against benchline implementations of similar algorithms.  It is almost trivial to run DDPG on Torcs using the openAI baselines package [https://github.com/openai/baselines].  I would have loved, for example, to see the effects of simply pre-training the DDPG actor on supervised data, vs. adding your mixture loss on the critic.  Using the baselines would have (maybe) made a very compelling graph showing DDPG, DDPG + actor pre-training, and then your complete method.

3) And finally, perhaps complementary to point 2), you really need to provide examples on more than one environment.  Each of these simulated environments has its own pathologies linked to determenism, reward structure, and other environment particularities.  Almost every algorithm I've seen published will often beat baselines on one environment and then fail to improve or even be wors on others, so it is important to at least run on a series of these.  Mujoco + AI Gym should make this really easy to do (for reference, I have no relatinship with OpenAI).  Running at least cartpole (which is a very well understood control task), and then perhaps reacher, swimmer, half-cheetah etc. using a known contoller as your behavior policy (behavior policy is a good term for your data-generating policy.)

4) In terms of state of the art you are very close to Todd Hester et. al's paper on imitation learning, and although you cite it, you should contrast your approach more clearly with the one in that paper.  Please also have a look at some more recent work my Matej Vecerik, Todd Hester & Jon Scholz: 'Leveraging Demonstrations for Deep Reinforcement Learning on Robotics Problems with Sparse Rewards' for an approach that is pretty similar to yours.

Overall I think your intuitions and ideas are good, but the paper does not do a good enough job justifying empirically that your approach provides any advantages over existing methods.  The idea of pre-training the policy net has been tried before (although I can't find a published reference) and in my experience will help on certain problems, and hinder on others, primarily because the policy network is already 'overfit' somewhat to the expert, and may have a hard time moving to a more optimal space.  Because of this experience I would need more supporting evidence that your method actually generalizes to more than one RL environment.

---

> ### Author Response · Authors · 2018-01-05
> **Response to AnonReviewer1**
>
> First of all, many thanks for your response. We have addressed the points you’ve mentioned in order to improve the quality of the article.
>  Below you can find our responses on your questions and suggestions.
>
> Considering the first point, thank you for mentioning this, in the new version of the paper we have changed the notation in order to improve the overall quality of presentation as you could see throughout all the text. We have also amended the text in a bid to improve the overall quality of the text.
>
> On the second and third points, we completely agree that we need some further tests. And it is greatly appreciated that you could even show us the way to conduct these experiments with OpenAI package. We have carried out such experiments on MuJoCo scenarios and presented them in the Appendix B.
>
> Considering the fourth point, we’ve read the paper by Matej Vecerik et al., it is indeed somewhat similar (but not the same) to what we propose. We have also amended the introduction to reflect that the authors of this paper are proposing similar ideas and contrast them to the ours. It is also remarkable that they are also thinking of applying it online to real-world scenario. However, their approach is different methodologically, as the authors of the paper are injecting the data into the replay buffer while we are regularising the Q-function. Also, it differs in terms of the application: the authors do not aim to use it for video and focus on robotic applications in real world.

---

### Official Review · AnonReviewer2 · 2017-11-26
**Benefit of RL portion is unclear**

**Rating:** 5
**Confidence:** 3

**Review:**

This paper proposes to combine reinforcement learning with supervised learning to speed up learning. Unlike their claim in the paper, the idea of combining supervised and RL is not new. A good example of this is a supervised actor-critic by Barto (2004). I think even alphaGo uses some form of supervision. However, if I understand correctly, it seems that combining supervision of RL at a later fine-tuning phase by considering supervision as a regularization term is an interesting idea that seems novel.

Having the luxury of some supervised episodes is of course useful. The first step of building a supervised initial model looks straight forward. The next step of the algorithm is less easy to follow, and presentation of the ideas could be much better. This part of the paper leaves me already with many questions such as why is it essential to consider only a deterministic case and also to consider greedy optimization? Doesn’t this prevent exploration? What are the network parameters (e.g. size of layers) etc. I am not sure I could redo the work from the provided information.

Overall, it is unclear to me what the advantage of the algorithm is over pure supervised learning, and I don’t think a compelling case has been made. Since the influence of the supervision is increased by increasing alpha, it can be expected that results should be better for increasing alpha. The results seem to indicate that an intermediate level of alpha is best, though I would even question the statistical significance by looking at the curves in Figure 3. Also, what is the epoch number, and why is this 1 for alpha=0? If the combination of supervised learning with RL is better, than this should be clearly stated. Some argument is made that pure supervision is overfitting, but would one then not simply add some other regularizer?

The presentation could also be improved with some language edits. Several articles are wrongly placed and even some meaning is unclear. For example, the phrase “continuous input sequence” does not make sense; maybe you mean “input sequence of real valued quantities”.

In summary, while the paper contains some good ideas, I certainly think it needs more work to make a clear case for this method.

---

> ### Author Response · Authors · 2018-01-05
> **Response to AnonReviewer2**
>
> Thank you so much for these valuable comments. We have carefully considered them in order to improve the contents of the paper.
> Below you can see our comments on your questions:
>
> “ This paper proposes to combine reinforcement learning with supervised learning to speed up learning. Unlike their claim in the paper, the idea of combining supervised and RL is not new. A good example of this is a supervised actor-critic by Barto (2004). I think even alphaGo uses some form of supervision. However, if I understand correctly, it seems that combining supervision of RL at a later fine-tuning phase by considering supervision as a regularization term is an interesting idea that seems novel.”
> Thank you for mentioning this previous works. We have cited Rosenstein and Barto in our new revision, and amended our claims (see the very end of introduction).
>
>
> “Having the luxury of some supervised episodes is of course useful. The first step of building a supervised initial model looks straight forward. The next step of the algorithm is less easy to follow, and presentation of the ideas could be much better. This part of the paper leaves me already with many questions such as why is it essential to consider only a deterministic case and also to consider greedy optimization? Doesn’t this prevent exploration? What are the network parameters (e.g. size of layers) etc. I am not sure I could redo the work from the provided information.”
> The greedy optimisation method was chosen in order to meet the requirements of real-time policy testing (first line in the while loop in the algorithm). We believe that the practical necessity of minimising the difference between the measurements per second rate in testing and training scenarios is vital for real world control scenarios as, unlike the gym environment, it would not be possible to wait for the training procedures. We’ve made the amendments in the discussion before the algorithm in order to highlight this issue. Considering the exploration prevention — we believe that it is a very valid case to explore, and we are now working on exploring this case in different reinforcement learning algorithms beyond the strategies described in this paper.
> Considering the network parameters — again, we completely agree with you that it was an omission, which we have corrected in this version (see updated Appendix A).
>
>
> “Overall, it is unclear to me what the advantage of the algorithm is over pure supervised learning, and I don’t think a compelling case has been made. Since the influence of the supervision is increased by increasing alpha, it can be expected that results should be better for increasing alpha. The results seem to indicate that an intermediate level of alpha is best, though I would even question the statistical significance by looking at the curves in Figure 3. Also, what is the epoch number, and why is this 1 for alpha=0? If the combination of supervised learning with RL is better, than this should be clearly stated. Some argument is made that pure supervision is overfitting, but would one then not simply add some other regularizer? ”
> Thank you for mentioning the error with epoch number. We’ve amended the algorithm in order to reflect our view that the epochs are counted from one, not zero, as it is in the rest of the text. We believe that this may be considered as a special kind of regulariser which is explicitly aimed to maximising the discounted reward.  And in order to further analyse the mutual impact and statistical significance of the improvement of the proposed method, we have made some additional tests with MuJoCo environment which we have presented in Appendix B. We show that in some cases (Hopper) the proposed regularisation could overcome not only the reinforcement learning algorithm but also the reference actor.
>
>
> “The presentation could also be improved with some language edits. Several articles are wrongly placed and even some meaning is unclear. For example, the phrase “continuous input sequence” does not make sense; maybe you mean “input sequence of real valued quantities”.
> We’ve made some amendments in the text in order to improve the presentation as you can see throughout the updated text of the paper. Many thanks for noting this.

---

### Official Review · AnonReviewer3 · 2017-11-27
**Unconvincing results**

**Rating:** 3
**Confidence:** 4

**Review:**


The paper was fairly easy to follow, but I would not say it was well written. These are minor annoyances; there were some typos and a strange citation format. There is nothing wrong with the fundamental idea itself, but given the experimental results it just is not clear that it is working.

The bot performance significantly better than the fully trained agent. This leads to a few questions:

1. What was the performance of the "regression policy", that was learned during the supervised pretraining phase?
2. Given enough time would the basic RL agent reach similar performance? (Guessing no...) Why not?
3. Considering the results of Figure 3 (right) shouldn't the conclusion be that the RL portion is essentially contributing nothing?

Pros:
The regularization of the Q-values w.r.t. the policy of another agent is interesting

Cons:
Not very well setup experiments
Performance is lower than you would expect just using supervised training
Not clear what parts are working and what parts are not

---

> ### Author Response · Authors · 2018-01-05
> **Response to AnonReviewer3**
>
> Thank you for these very meaningful comments. In the following paragraphs we explain the amendments we’ve made in order to address the raised issues.
>
> Considering the first point, ‘What was the performance of the "regression policy", that was learned during the supervised pretraining phase?’, we’ve amended the text of the article to explain that, according to Algorithm 1, the pretraining stage performance is evaluated during the first epoch. Therefore the points for the stage one in the graphs in Figure 3 show the performance of the retrained stage. We have put the additional explanations to the section 3.2.
>
> For the second point, “Given enough time would the basic RL agent reach similar performance? (Guessing no...) Why not?” In order to make the necessary assessments for this point, we have carried out the additional experiments in Appendix B on MuJoCo tasks, which confirm that while usually it is limited by the performance of the RL method in some cases pretraining allows to go even beyond the capabilities of both the ‘pure’ RL method and the supervised model performance (see Figure 6, Hopper scenario). But our claim is that by pretraining and supervised learning assistance we minimise the time of ‘nonsense’ control signals with extremely low rewards in order to enable the real-time training scenarios (with potential applications to real world environments).
>
> Considering the third point, ‘Considering the results of Figure 3 (right) shouldn't the conclusion be that the RL portion is essentially contributing nothing?’, despite the reasonably bad performance of the RL portion on this task, the combination of reinforcement and supervised learning still provides better results in terms of both maximum and average rewards. But as we totally agree with you that this evidence in the original paper was not sufficient, we hope the additional experiments on MuJoCo tasks would strengthen this point.

---

### Author Response · Authors · 2018-01-05
**Summary of the amendments**

Many thanks for the very useful comments from all the reviewers. We have taken them into account with the following list of amendments:
- We have added a new Appendix B, describing the results of the experiments on MuJoCo tasks. Reflecting these changes, we have also added the references on OpenAI baselines and MuJoCo publications.
- Throughout the description of the method, we have made the notation closer to the one used in many of the papers within the community (a is replaced by \pi, and where it was possible, we have removed parameterisation Theta_pi, which was cluttering the notation).
- We have slightly amended the claims (in abstract and the introduction) in order to address the comments from the reviewers. It includes: stating in the introduction that the combination of reinforcement and supervised learning did exist before but not in the problem statement of supervised regularisation for the optimisation problem; adding the information about the previous works in supervised actor-critic by Barto (2004), and also Matej Vecerik, Todd Hester & Jon Scholz: 'Leveraging Demonstrations for Deep Reinforcement Learning on Robotics Problems with Sparse Rewards’. We have also contrasted in the introduction the differences between those approaches and the proposed one.
- We have also repeatedly proofread the text in order to remove ambiguities, including those found by the reviewers ( “continuous input subsequence” -> “subsequences of real valued quantities”).  Also we have stated explicitly for the Algorithm 1 that “the 0-th epoch's testing episodes reflect the performance of the model with supervised pretraining.”
- We have changed the alignment of some figures (notably Figure 4 and 5) in order to  improve presentation
- We have added the network parameters (sizes of the layers) to Appendix A to ensure repeatability of the experiments.

---

### Decision · Program_Chairs · 2018-01-29
**ICLR 2018 Conference Acceptance Decision**

**Decision:**

Reject

**Comment:**

The proposed method combines supervised pretraining given some expert data and further uses the supervision to regularize the Q-updates to prevent the agent from exploring 'nonsense' directions. There a significant problems with the paper: the approach is not novel, the assumption of large amounts of expert data is problematic, and the claim of vastly accelerated learning is not supported empirically, either in the main paper or in the additional mujoco experiments added in the appendix.